# Effective and Efficient Delivery of Genome-Based Testing-What Conditions Are Necessary for Health System Readiness?

**DOI:** 10.3390/healthcare10102086

**Published:** 2022-10-19

**Authors:** Don Husereau, Lotte Steuten, Vivek Muthu, David M. Thomas, Daryl S. Spinner, Craig Ivany, Michael Mengel, Brandon Sheffield, Stephen Yip, Philip Jacobs, Terrence Sullivan

**Affiliations:** 1School of Epidemiology and Public Health, University of Ottawa, Ottawa, ON K1G 5Z3, Canada; 2Office of Health Economics, London SE1 2HB, UK; 3City Health Economics Centre (CHEC), City University of London, London EC1V 0HB, UK; 4Marivek Healthcare Consulting, Epsom KT18 7PF, UK; 5Garvan Institute of Medical Research, Sydney, NSW 2010, Australia; 6Omico, Sydney, NSW 2010, Australia; 7Menarini Silicon Biosystems Inc., Huntingdon Valley, PA 19006, USA; 8Provincial Health Services Authority, Vancouver, BC V5Z 1G1, Canada; 9Department of Laboratory Medicine & Pathology, University of Alberta, Edmonton, AB T6G 2S2, Canada; 10William Osler Health System, Brampton, ON L6R 3J7, Canada; 11Department of Pathology and Laboratory Medicine, Faculty of Medicine, University of British Columbia, Vancouver, BC V6T 1Z7, Canada; 12Faculty of Medicine and Dentistry, University of Alberta, Edmonton, AB T6G 2R3, Canada; 13Institute of Health Policy, Management and Evaluation, University of Toronto, Toronto, ON M5T 3M6, Canada; 14Gerald Bronfman Department of Oncology, McGill University, Montreal, QC H4A 3T2, Canada

**Keywords:** diagnostic molecular pathology, genetic testing, diagnostic services, technology assessment, biomedical, genetic services, financial support, clinical governance, health technology, health care innovation

## Abstract

Health systems internationally must prepare for a future of genetic/genomic testing to inform healthcare decision-making while creating research opportunities. High functioning testing services will require additional considerations and health system conditions beyond traditional diagnostic testing. Based on a literature review of good practices, key informant interviews, and expert discussion, this article attempts to synthesize what conditions are necessary, and what good practice may look like. It is intended to aid policymakers and others designing future systems of genome-based care and care prevention. These conditions include creating communities of practice and healthcare system networks; resource planning; across-region informatics; having a clear entry/exit point for innovation; evaluative function(s); concentrated or coordinated service models; mechanisms for awareness and care navigation; integrating innovation and healthcare delivery functions; and revisiting approaches to financing, education and training, regulation, and data privacy and security. The list of conditions we propose was developed with an emphasis on describing conditions that would be applicable to any healthcare system, regardless of capacity, organizational structure, financing, population characteristics, standardization of care processes, or underlying culture.

## 1. Introduction

Clinical decisions are increasingly informed by biomarker-defined subsets of patients, forming the basis of precision medicine. The measurement of laboratory-based biomarkers may serve one or more purposes for patients and healthcare providers—including identifying who has disease or may develop future disease, monitoring health, monitoring response(s) to therapy, predicting who may most benefit or be harmed by therapy, or predicting how disease is likely to progress [1]. Increasingly, advanced tests are being used. These involve measures of the expression, function and regulation of genes through a direct examination of genes (through cytogenetics or various genomic tools), or their protein products (for example by immunohistochemistry). These techniques have already become a hallmark of decision-making in oncology, given cancer is a genetic disease and may be amenable to targeted therapy. They are also increasingly playing a role in the identification of disease, with comprehensive genomic testing approaches demonstrating much higher diagnostic yields than conventional approaches. A prospective study of 103 children with suspected underlying genetic disorders led to 18 new diagnoses of disease when using whole-genome sequencing compared to conventional approaches to diagnosis (i.e., targeted gene sequencing) [2].

In this rapidly developing field, healthcare system administrators are faced with decisions regarding what biomarkers and technologies to adopt. These include different technical platforms (e.g., single-gene, multi-gene, whole-exome, and whole-genome sequencing and expression analysis); modalities (tissue, saliva, blood, or urine-based sampling); location (laboratory-based or delivered at point-of-care); provenance (commercially available in vitro diagnostic tests and services versus in-house/laboratory developed tests); and timing and sequencing of tests. All of these factors affect clinical utility, including costs and patient outcomes, and broader health system goals such as caregiver and patient experiences. They will also influence how care is delivered.

Realising benefits from genome-based testing requires expenditure on equipment, reagents and informatic tools for their mainstream delivery, and creates added challenges for healthcare systems and their stewards. These challenges include, but are not limited to: management costs associated with development of new protocols, implementation, validation and licensing; the need to revisit clinical and referral pathways, scopes of practice, and care protocols; the need to revisit use of associated resources, including upstream tissue sampling and downstream use of targeted therapies; the need to define minimally acceptable technical standards to allow for collection of samples; the need to revisit associated human resource requirements, such as laboratory technologists/technicians, bioinformaticians, clinical geneticists, and genetic counsellors; enhanced data systems in evaluation and reporting, that also consider privacy and security. These challenges combined speak to a need to revisit historical models of technology management, including the governance, administration, and financing of laboratory services.

Unlike traditional healthcare technologies, genome-based biomarker testing also provides research opportunities beyond healthcare decision making, and may lead to discoveries about the nature of disease or effectiveness of current and future therapies. These opportunities are being developed in ‘real time’, within routine clinical use, thus blurring the boundaries between research and standard practice. For example, tests may qualify patients for clinical trials, which has important clinical, scientific and economic benefits [3]. The value of integrating research into clinical practice was underlined during the COVID-19 pandemic, where rapid research-led development and implementation of testing capacity was essential for both monitoring the pandemic, and the development of preventive and therapeutic strategies. Yet, the value of research in healthcare is not integrated into health technology assessment in most single payer systems [4].

Given the potential for exponential growth of new tests and test approaches, and the complexity of introducing them, health system planners preparing for a future of genome-based biomarker testing need to grapple with how these services can be delivered effectively and efficiently. As the structure, remit and organization of healthcare systems (and the laboratory functions within them) vary, there are likely to be no one-size-fits-all solution; however, some necessary conditions will be required to manage these technologies in a way that benefits patients and is sustainable.

The purpose of this report is to describe the conditions necessary for policymakers and health system planners to enable a state-of-the-art testing service that includes genome-based testing for acquired and heritable diseases as well as risk assessment to support preventive and public health interventions. The list of conditions we propose was developed with an emphasis on describing conditions that would be applicable to any healthcare system, regardless of capacity, organizational structure, financing, population characteristics, standardization of care processes, or underlying culture [5].

## 2. Materials and Methods

The conditions identified in this report were developed through a mixed-methods approach. A narrative literature review was conducted based on a purposive sample of commercially published and grey literature. Searches (See Appendix A for search strategy) were performed by a medical librarian specialist and relevant information was identified by a single reviewer (DH). In parallel, conditions were identified using a conventional content approach and based on semi-structured interviews (*n* = 18; 30–60 min) with key informants and performed from a constructivist point of view. All interviews were conducted by DH with a purposive sample of experts including several of the authors (VM, DMT, DSS, CI, MM, BSS) as well as representatives from pharmaceutical (*n* = 6) and diagnostic (*n* = 1) companies. Informants were chosen based on differing expertise and geographic location with some (*n* = 4) having previously worked with the author.

Interviews were conducted via a recorded video conference call using an interview guide, and all participants approached agreed to be interviewed. Summary notes from transcripts were shared back with participants for verification (member checking). An informal identification of concepts was conducted by one author (DH), and categorized into themes/conditions.

A preliminary list of conditions identified through interview and the literature review was then circulated back to all authors for feedback and a moderated face-to-face discussion. “Effective and efficient” delivery of genome-based diagnostics from testing was defined as one that would most satisfy the “quadruple aim” of reducing per capita costs while improving population health outcomes, patient and caregiver experiences, and provider experiences [6]. These conditions are explained, and grouped according to these aims, and with examples in the next section.

## 3. Results

Interviewees (Table 1) described a number of challenges in achieving the quadruple aim of healthcare within current approached to the implementation and management of genomic testing. These largely related to care interruptions or wait times due to a number of underlying factors (resources and finance planning; education; informatics; and an unclear process for onboarding tests). Other challenges included inappropriate identification of patients and family members; inequitable care delivery; uncoordinated, inconsistent, inappropriate or duplicative care; and inefficient, low-value care. These are grouped by theme in Table 2.

From the challenges identified, an initial list of 14 conditions were identified and pared down to 12. These conditions for testing have been characterized as being part of infrastructure and planning; operations; or the general, health care environment. The sections below provide further elaboration of these, with examples. A summary of conditions, along with a description of issues they are intended to address, and the goals and description of good practice are in Table 3.

### 3.1. Infrastructure and Planning

#### 3.1.1. Creating Communities of Practice and Healthcare System Networks

A well-established network can serve as a basis for deliberation about what tests must have priority, how tests may be valued, what care standards should be in place, what resources will be necessary, how care can be monitored, and other necessary collective judgements that may vary geographically and over time. There will also need to be broad agreement on the use of shared resources, such as biobanks and reporting standards. In regions with more dispersed delivery of care, efforts to create networks may require tiering: first, there is a need to establish across-region consortia to establish wider care standards, shared informational resources, educational standards, and ensure the equitable delivery of services [23]; secondly, there is a need to create strong intraregional consortia for fully integrated delivery of services (i.e., local communities of practice or “collaborative communities” [24]). 

Testing is a complex intervention [25] that relies on the timing, expertise, and behaviour across multiple stakeholders for its effective delivery. At its core is a community of practice that includes laboratory leaders and healthcare providers who will have the greatest impact on multidisciplinary decisions in regards to testing, including how and under what conditions a test should be delivered [26]. Broader members of the community are those who will be impacted by the consideration and implementation of new testing. These include the patients, administrators, IT professionals, implementation and genome scientists, public and private sector innovators and others (scientists, legal and ethics experts, professional organizations, bioethicists, regulators). As these groups may not be connected through an organizational structure, strong networks with clear communication between individuals and programs are required for effective implementation and good decision-making [24].

Many international jurisdictions have already established networks through translational research initiatives [27]. In pluralistic or federated health systems such as Sweden and Canada there has been an emphasis on “bottom-up” approaches to creating regional capacity that foster the building up of self-selected organizations aligned with a core set of goals. Some federated jurisdictions, such as Australia, have taken a further “top-down” approach to creating networks after mapping jurisdiction-wide capacity. The Australian Genomics Health Alliance, for example, is an attempt to accelerate and evaluate the application of genomic testing in healthcare. It is a “collaborative research partnership across more than 80 diagnostic laboratories, clinical genetics services, and research and academic institutions” [7].

In more centralized healthcare systems such as England’s National Health Service (NHS England), where higher level coordination already exists, the emphasis has been on regional care coordination. Collaboration in England has been facilitated by creating the NHS England Genomic Medicine Service Alliance, an effort to bring together Genomic Laboratory Hubs together with “clinical genetic services inclusive of genomic counsellors, provider organisations across the care continiuum [sic] and with Primary Care Networks, Cancer Alliances, research and academia and patients and public representatives” [28]. In parallel, a consortium of academic researchers (a community of approved researchers with access to the Genomics England Research Environment) was also created as a mechanism of reaping benefits from scientific spillovers from genomic information.

#### 3.1.2. Resource Planning

Resource management and planning for expected impacts on time, people, facilities, equipment, supplies, and information technology is an essential activity in any health system. However, the rapid rate of change of underlying technology and the need for specialized human resources including those involved in tissue sampling (e.g., biopsy), analysis (laboratory technologists/technicians, bioinformatic) and post-test counselling (counsellors and other specialized training) necessitates long-term capital and human resource planning. Workforce planning will need to consider the training and credentialling of highly specialized resources involved with testing. It may also need to consider a plausible range of scenarios of what services are required and the roles and responsibilities of those involved [29,30]. For example, the US Government Accountability Office conducted a study forecasting a future shortfall of genetic counsellors and medical geneticists in general, and by geographic region [8]. Advances in searchable genomic databases to support clinical management, alternative models of service delivery, and centralized delivery of services could greatly reduce the need for these highly specialized human resource requirements in coming years [23,31,32].

Resource planning for the coming era of genomic medicine will require health system planners to revisit traditional funding formulas. Laboratory funding based on volume or a “per-test” approach may not incent its use as it ignores efficiencies that could be realized with changes in approach to testing type (e.g., multigene assay versus single gene approaches [33]), modality (e.g., reflex testing or upfront testing versus ordered testing or sequential testing [34]) or test timing [35,36]. Additional bioinformatics and technologist/technician resources also require consideration. In the UK, for example, the Department of Health & Social Care committed “£4 billion over a five-year period (2016–21) in digital technology, systems and infrastructure, to provide the health and care system with the digital capability and capacity it needs ….” [9].

#### 3.1.3. Informatics

Traditionally, laboratory information management emphasizes informatics as a tool for sample tracking and communicating results. For genome-based testing, informatics is also essential for test development, interpretation, and clinical decision support [37,38]. Ensuring adequate integration of test results into electronic health records will also provide a key resource for real-world monitoring, disease management, quality assessment and assurance, and financing [39]. Integration of laboratory information systems with electronic health record systems is also needed to reduce duplication of testing and as a basis for care coordination across health professionals or organizations. An increasingly important aspect of genome-based testing is the ability not only to share information within a healthcare system, but also to share and access data from other informational sources, including external databases, peer-reviewed literature and other healthcare systems.

Histopathology represents a major portion of laboratory medicine and involves images interpreted by human physicians. This area has existed for over a century as an unquantifiable practice within medicine. In the current digital era, digitization of stained images represents a major advance in the practice of personalized medicine. As informatics capability continues to expand, health systems may plan for the integration of digital histomorphologic data and its ongoing analysis into genomics and personalized medicine [40].

### 3.2. Operations

#### 3.2.1. Entry/Exit Point for Innovation

The rate of proposals for new tests and testing modalities necessitates a clear process for the managed adoption and obsolescence of tests [41]. A single point of entry for considering new tests using an application procedure coupled with an evaluation process and formulary is one increasingly used approach that allows multiple stakeholders to engage with the healthcare system [42]. It can also reduce unnecessary testing while providing a strong signal to public and private sector innovators regarding when and under what conditions tests will be adopted [43]. This approach is used in the Canadian provinces of Alberta and Quebec, where an intake form requesting a new test (councils, strategic clinical networks, physicians, patients, innovators or the public) will lead to an evidence-assessment and recommendation before a test is placed on a public formulary.

Given the rate of technologic development, healthcare systems will also need to grapple with timeliness, i.e., how long to adopt tests and when tests should be reassessed. New test adoption is a healthcare challenge, as many tests must be considered in the context of other interventions, such as their use as companion diagnostics for new drugs. Any decision to replace or revise an existing test, such as expanding a multigene panel, must consider the balance between patient and healthcare provider unmet need with the inevitable disruption to care protocols, and the speed at which new tests can be replaced. NHS England, for example, has announced its decision to revisit tests annually, and considering the coordinated replacement of older tests with new and emerging approaches, including considering where evidence still needs to be collected to validate the benefit of moving to [whole-genome sequencing], and identifying where alternative genomic diagnostics, such as gene panels or microarrays, will continue to be needed [9].

#### 3.2.2. Evaluative Function

Many health systems worldwide have adopted evaluation frameworks for testing based on the analytic validity, clinical validity, clinical utility, and ethical, legal, and social implications (ACCE) approach [44]. While this can be seen as a starting point for evaluation of tests from a clinical standpoint, the ACCE approach does not consider “context-related evaluation dimensions (delivery models, economic evaluation, and organizational aspects)” that will be of interest to policymaking and are a standard part of health technology assessment (HTA) processes [45]. Even using the ACCE framework, it has been argued that ‘clinical utility’ is poorly defined; definitions “may focus on a test’s ability to produce a diagnosis, broader definitions of clinical utility consider health and non-health related, familial and societal outcomes”. Expanded notions of utility, that consider the wider impact of a test result on not only the individual patient, but their families and broader society, may be required to capture benefits to society [46,47].

Many issues related to testing, including availability of evidence and context-driven performance are part of a broader suite of limitations faced by administrators when evaluating diagnostic tests and medical devices [48]. As such, the timing and complexity of traditional approaches to HTA must be balanced against patient need, in an environment where test utility and cost is dynamic. Literature-based approaches to estimating test utility may be limited by enhancements to technology, learning curves [49] or implementation characteristics that affect performance [50]. This strongly suggests the recommended use of both pre-market and post-market data to capture impact of learning curve on outcomes [4] as well as a consideration of costs of implementation when assessing value [51]. Furthermore, and in keeping with key principles for HTA [10], decisions regarding access to testing must be made in a timely manner. Both of these issues are addressed in an evaluative framework for genetic testing developed for the US Department of Defense which recognized the practical need to triage adoption decisions based level on urgency through the use of rapid review and real-world evaluation of new tests [11].

Traditional assumptions and approaches underlying the economic evaluation of decisions for drugs are also challenging to apply [52]. These include the constancy of marginal benefits and costs, and the divisibility of tests provided [53,54]. Payers should expect the marginal costs of adding new tests to a panel or going to a whole exome or genome approach to be quite small relative to other factors such as patient selection, level and type of implementation (education of providers, equipment, geographic distribution) as well as downstream costs (e.g., use of targeted therapies) [55].

#### 3.2.3. Service Models

Like many other forms of production, good practices in organizing health services needs to consider the degree to which the configuration of delivery is concentrated or dispersed. Dispersed arrangements are more attractive when unit costs do not benefit from economies of scale—such as with primary care and community pharmacy services—and coupled with the need for geographic reach [30,32]. In contrast, economies of scale from advanced testing, coupled with the need for a high degree of standardization and accountability suggest genome-based testing will benefit from a more concentrated model of service delivery.

A further consideration will be to what degree a more concentrated model can be delivered. Service models must first consider care pathways and requisition authority for testing. Models include requisitions by geneticists, primary care practitioners, medical specialists, program-based (such as newborn screening) or direct-to-patient and will depend on the purpose of testing [56]. Genetic testing may be required by independent healthcare programs that are uniquely organized, such as prenatal, pediatric, infectious disease, psychiatry, primary care, and oncology. In some cases, a single test (e.g., BRCA) may be used to assess future risk of disease, prognosis of disease, or predict response to treatment could have different clinical applications, referral pathways, and healthcare system value.

Point-of-care testing (POC) technology is also increasingly available, which can expedite decision-making but also challenges a more centralized model of care, and the standardization and accountability that comes with it. POC devices do not negate the need for quality control, external quality assessment, provider training, and data sharing associated with testing. POC tests also further highlight the need for technology adoption decisions that consider an entire community of practice—differences in speed of test results and analytic characteristics will have a downstream impact on patient and healthcare provider experiences.

Care coordination may be greatly facilitated by already-centralized healthcare environments. The NHS England was able to reorganize its existing capacity in 2018, creating a Genomic Medicine Service through its Genomic Laboratory Hubs, each hosted by an acute NHS trust and designated a geographic region for coverage [12]. Similarly, the US Department of Veteran’s Affairs has leveraged its existing capacity to deliver genetic testing through its oncology program and dedicated service centres across the US toward non-oncologic indications for testing [13].

In pluralistic or market-based healthcare systems, coordination of care across disparate organizations is facilitated through the use of care standards linked to incentives. Israel, for example, has decided to create regional capacity to deliver comprehensive genomic profiling for non-small cell lung cancer by allowing its separate health management organizations (Kupot Holim) to use their own validated testing approaches, in accordance with specific conditions. Spain, has similarly provided overarching guidance to its autonomous health regions regarding the principles that underly the delivery of genetic tests [57]. Even when designed with best intentions, these approaches may still lead to regional variation and concerns regarding inequity of access. The Spanish Minister of Health has recently announced further efforts will be made to make country-wide access to testing more consistent [58].

#### 3.2.4. Awareness and Care Navigation

Even with necessary test infrastructure and accessibility, healthcare systems must consider how to communicate to patients and healthcare providers what tests are available and to who and how these are paid for. Published test formularies are a starting point for communicating what tests are available and how they can be accessed. In the Canadian province of Alberta, the addition of tests to a formulary had a stated goal of “streamlining processes by reducing variation in testing and improving healthcare provider and patient access to appropriate, equitable and sustainable laboratory test information [59]”. In countries with pluralistic healthcare systems and lacking a common directory, other information can be provided to help care navigation. In France, where testing is more variable across regions, lists of different laboratory sites with contact information are provided [14]. Similarly, the US NIH has developed a test registry, which “contains information about laboratories and the tests they offer but does not contain or gather information on genetic test results” [60].

### 3.3. Healthcare Environment

#### 3.3.1. Integration of Innovation and Healthcare Delivery Functions

Given the rapid future pace of the introduction of new tests, coupled with the potential research benefits associated with testing, healthcare systems will need to consider how the delivery of testing or scientific discovery alongside testing for healthcare decision-making will be coordinated. This is an inherent challenge with exome- or genome-wide sequencing, which will invariably reveal genetic variants of strong therapeutic, prognostic, or diagnostic significance alongside those lacking evidence [61]. Investigational tests can play an important role in patient care: qualifying patients for clinical trial enrollment, as well as other research endeavors that further understanding of disease. European guidelines have addressed this by suggesting the distinction is clear when reporting results [26]. In Ontario, Canada, for example, reflex testing for newly diagnosed cases of NSCLC (adenocarcinoma/non-squamous) is performed using a panel consisting of more well-established biomarkers. Targeted treatments are currently available for some of the genes tested (e.g., EGFR, ALK, KRAS), but not for others (e.g., FGFR1, SMARCA4, PIK3CA) [17].

Funding tests for “targetable” and “non-targetable” genes together is a pragmatic solution, and is also facilitated by massive parallel (“next-generation”) sequencing where additional tests can be added to an assay at negligible cost. It can also allow health systems to revisit testing regimes with less frequency, and avoid significant change management costs. In practice, however, funding both “medically necessary” and “investigational” testing can create a significant conflict for existing insurer frameworks that use evidence and clinical consensus to determine what biomarkers should be funded [62,63]. This challenge led one commentator to ask: “do we redefine [testing] to fit the coverage and evidence framework, or do we redefine the coverage and evidence framework to fit [testing] [63]?”

Proposals to change coverage frameworks have been well described and are intended to address payer risk through performance-based payment or coverage with evidence development [63,64]. While a step forward, these solutions may still be difficult to implement in practice, given the inherent limitations of using real-world data to establish the clinical utility of testing [65]. Approaches to circumventing evidence challenges include the use of standardized outcome measures, cascade testing and data sharing through international consortia [66].

A separate solution is to create translational research programs that work in parallel with health systems, or ideally are fully embedded within learning healthcare systems [66]. Many of these already exist today, often facilitated through public-private sector partnerships, the majority intended to investigate normal genomic variation by sequencing healthy participants (i.e., biobanking) [27]. Some also have stated aims of drug discovery and integrating well-established and emerging tests into regular healthcare system delivery. The Australian government, for example, has created unique partnerships between government, industry, and academia to conduct clinical trials to establish the clinical utility of comprehensive genomic profiling in lung cancer [15], as well as a more recent announcement for rare disease [16].

#### 3.3.2. Financing Approach

The anticipated rate of entry of new tests also requires a nimble financing approach, allowing funds to be released for new tests once decisions to reimburse are made. This may require a shift in thinking for many insurers, who have historically allocated funding for laboratory services on an annual basis based on test volumes [64]. Unlike traditional tests, funding formulas for genetic testing must consider the need for additional human resources associated with development and proficiency testing [67]. Payment models for care may, in turn, drive laboratory utilization and require re-thinking [64,68].

In the US, the Centers for Medicare and Medicaid Services (CMS) have attempted to incentivize molecular diagnostic innovation by enabling manufacturer-set free pricing on the Medicare fee schedule for tests that meet specific Advanced Diagnostic Laboratory Test (ADLT) criteria. Qualifying tests must be covered under Medicare, provided by a single laboratory, and either (1) be FDA-cleared or approved, and/or (2) meet three specific criteria ensuring molecular diagnostic innovation [18].

Some insurers have also established funding for genetic biomarkers predicated on a “companion” diagnostic paradigm, releasing funds only when companion drugs are approved. Genome-based biomarkers, however, are increasingly used for multiple drugs or therapeutic decisions, including decisions not to use older drugs, and to shorten diagnostic odysseys for conditions with no specific drug therapies [69,70,71,72]. In some cases, jurisdictions have additionally relied on pharmaceutical companies or public sector research grants to fund these one-drug, one-test dyads. For public insurers, this inevitably creates a situation where public sector actors are dependent on the private sector (or others) for the delivery of public services, and yet public actors remain accountable to the public at large. This “private finance initiative” type of problem means testing health system priorities are dictated by who is paying, rather than unmet need, equity, or efficiency [73]. In addition to creating structural inefficiency, these arrangements may be disruptive if funds are quickly withdrawn: research grant funding may cease or move away with an investigator; or, a drug company may change its external funding policies. The same company may also reasonably not want to pay for tests that aid competition.

#### 3.3.3. Education and Training

Genetic testing through interprofessional teams distributed across centres and programs introduces new challenges for educating healthcare professionals when creating system-wide changes. Implementation new genome-based tests will change workflow, and necessitates training at the intersection of continuing professional development, knowledge transfer and quality improvement [32,49,74]. This in turn may require new approaches to teaching including workplace-based assessment and in situ simulation that address the many contextual requirements of testing that can change ultimately affect test performance, including “coordination of care, tissue procurement and handover, requisition and report design, clear workflow within and between services, automatic information exchange between electronic health systems, and improved communication, with fast feedback loops between health care practitioners [75]”.

The need for a significant level of education caused by a significant disruption to organization of services is reflected in the approach proposed by the Genomics Education Programme (GEP) in England. The Programme “routinely engages with the Medical Royal Colleges and actively participates in the NHS England and Academy of Medical Royal Colleges (AoMRC) Genomic Champions Group”. Among other areas of focus, the GEP plans to develop “genomic competencies for specialty training”, human resource planning, and providing supports for “curricula development and medical revalidation [9]”. Funding for each of the Genome Laboratory Hubs also considers the need for education and training. Some jurisdictions have even funded programs aimed at improving genomic knowledge in school-aged children [76]. 

#### 3.3.4. Regulation

While some health product regulators, such as the FDA, have begun to test claims of clinical validity for commercially available tests, these do not address the multitude of factors that ultimately contribute to test utility and cost-effectiveness. This has heightened the need for effective systems of regulation to address the numerous factors that contribute to test quality, including human resource qualifications and training, documentation of records, quality control processes, and proficiency testing [19,20,21]. Additional consideration must be given to the training, licensure, registration, and certification of genetic counsellors [77].

Most jurisdictions recognize that advancing the quality of testing requires regulatory standards that involve multiple stakeholders, as it is widely recognized unwanted variation in test performance is largely driven by steps taken before and after analysis [20]. Regulation is typically addressed through accreditation processes that conform with the International Organization for Standardization (ISO) including ISO 15,189 Medical Laboratories. Examples include regulation of clinical genetic testing through CLIA in the US and Canada, and through the National Association of Testing Authorities and the Royal College of Pathologists of Australia (NATA/RCPA) in Australia. In Europe, the CF Network, ERNDIM, GenQA (formally CEQA) and EMQN have more recently harmonized accreditation standards.

#### 3.3.5. Data Privacy and Security

The proper interpretation of disease-gene relationships, particularly for rare variants, require significant amounts of information including family histories and shared information across laboratories, both locally and internationally. The availability of partial or complete genomic information, however, may allow individuals to be identified. Testing also raises ethical issues, such as the duty to warn first-degree relatives who may have a high chance of carrying a disease-causing gene [61]. As such, data requirements associated with genetic testing raise privacy and security concerns that may require revisiting of historical legislation or policies.

Some of these challenges may be overcome through the adoption of technical solutions and data standards. Technical solutions include privacy-preserving solutions used in information systems, such as blockchain, to help avoid de-identification. Data sharing to community resources, such as the US National Library of Medicine’s ClinVar, is also strongly encouraged in international guidelines. Frameworks for data sharing have been bolstered by international efforts, such as the Global Alliance for Genomics & Health, which has created a Framework and “Core Elements for Responsible Data Sharing” [22]. The framework emphasizes maximizing data accessibility of data while minimizing harm to patients and others through a transparent and accountable system.

Data privacy and security concerns may also be addressed through education and training (see Section 3.3.3). A core competency framework developed by NHS Health Education England, for example, identified six areas of proficiency for those responsible for communicating test results. A part of the framework addresses appropriate communication of genomic results, including understanding “the implications of genomic testing for insurance, including the UK Code on Genetic Testing and Insurance” [9].

Some genetic tests may also require outsourcing, due to rarity or health system capacity. Genetic testing opens up the possibility of an output of unprocessed genetic data that requires interpretation from out-of-country providers. In these situations, health system administrators will also need to consider what information can or should lawfully be shared across borders.

## 4. Discussion

Our effort to capture necessary conditions for state-of-the-art genome-based diagnostics service is intended to aid those who must design policies and processes intended to capture the value of genome-based testing. While there is much focus on health technology assessment (HTA) and economic evaluation as a policy response to new technology, we would suggest appropriate management of health technology goes much further than HTA [78]. The conditions listed here reflect broader conditions of high performing health systems that have been previously described [79,80,81,82]; these include the need to consider accreditation, regulation, provider training, care coordination, health information technology, evidence-based policy, and financing as a means to reduce inequity, improve care recipient and provider satisfaction, while moderating the rate of expenditure.

While many of the conditions for optimal delivery of care could be applied to other disruptive technologies, the key findings of our review suggest there are some conditions and good practices that will be strongly emphasized in a high-performing genome-based testing service. These include across-region informatics associated with testing, a framework that addresses privacy and security concerns from genetic testing, as well as integration of an innovation and healthcare delivery function through private public sector partnerships or the sanctioned use of investigational technology in mainstream healthcare. Improvements in these areas are significant challenges but necessary ones for a future of learning health systems [83].

## 5. Conclusions

We have identified 12 necessary conditions required for policymakers and health care system planners to achieve optimal experiences for care providers, patients and caregiver while achieving better outcomes and minimizing per capita health care costs in the coming era of genomic medicine. As these conditions have been identified through a comprehensive literature review and key informant interviews with international experts, they should be applicable to any healthcare system, regardless of capacity, organizational structure, financing, population characteristics, standardization of care processes, or underlying culture.

These conditions also reflect the multifaceted nature of laboratory technology management as well as the need for additional considerations beyond traditional laboratory technology. As genome-based testing becomes more prevalent in coming years, we hope these conditions and accompanying examples of good practice internationally provide some initial guidance for those who will need to redesign healthcare systems to optimize care.

## Figures and Tables

**Table 1 healthcare-10-02086-t001:** Interviewee characteristics.

Participants (*n* = 18)	N, (%)
Female	6 (33)
Primary role	
Physician/lab leader	4 (22)
Health care administrator	2 (11)
Health services expert/health economist	4 (22)
Patient representative	1 (6)
Private sector representative	7 (39)
Work environment	
Public sector	11(61)
Private sector	7 (39)
Location	
Canada	14 (78)
United States	2 (11)
Europe	1 (6)
Other	1 (6)

**Table 2 healthcare-10-02086-t002:** Thematic analysis of challenges identified along with corresponding enabling conditions.

Quadruple Aim Domain *	Challenge/Theme Identified	Key Informant Quote	Potential Solution/Condition(s) to Help Address
Work life of care providers	Care interruptions and wait times	“It is a challenge to connect different streams of planning. A nimble lab service is highly dependent on integrated lab systems and capital planning. Workforce planning and education is also critical” “There is a need to triage the urgency based on the test application and clarity about the prerequisite level of evidence to apply”	• Resource planning • Financing approach • Education and training • Informatics • Evaluative function • Entry/exit point for innovation
Patient and caregiver experiences	Inappropriate identification of patients and family members	“There is a need for standards around governance, security and patient consent, what we can use the data for etc. Rule around commercial interests in data need to be in place [as well as] some consideration of investing in the laboratory function independently of therapeutic application”	• Service models • Awareness and care navigation
	Inequitable care delivery	“Legislation plays an important role as well. The Acts give different provinces different levels of influence over care coordination”	• Regulation
Health of populations	Uncoordinated, inconsistent, inappropriate or duplicative care	“There is a need to have clear understanding of what the care pathway is and an aligned community of practice”	• Integration of innovation and healthcare delivery • Creating communities of practice and healthcare system networks
Per capita costs of healthcare	Inefficient, low-value care	“So then we invest in standards, outcome measures, quality measures etc. along with a process. You can’t be too prescriptive because of the wide utility of testing” “Information is also valuable and must be valued. Currently information is generated for medico-legal purposes and yet it could be generated to generate revenue and lower care costs”	• Integration of innovation and healthcare delivery

* Challenges and themes span across some quadruple aim domains, so rows are deliberately misaligned.

**Table 3 healthcare-10-02086-t003:** Enabling conditions for state-of-the-art delivery of genome-based testing.

	Issue	Goal	Description of Good Practice	Policy Example
**Infrastructure**				
Creating communities of practice and healthcare system networks	Inequitable care delivery	▪ Broad stakeholder agreement on appropriate use ▪ Equitable care	▪ Engagement across all stakeholders	The Australian Genomics Health Alliance, for example, is an attempt to accelerate and evaluate the application of genomic testing in healthcare. It is a “collaborative research partnership across more than 80 diagnostic laboratories, clinical genetics services, and research and academic institutions” [7].
Resource planning	Care interruptions, wait times or unsustainable care	▪ Sustainable care delivery	▪ Frequent (e.g., 1–3 years) reassessment ▪ Available to all healthcare stakeholders	The US Government Accountability Office conducted a study forecasting a future shortfall of genetic counsellors and medical geneticists in general, and by geographic region [8].
Informatics	Uncoordinated or duplicative care, inconsistent test development, poor information for evaluation	▪ Care coordination ▪ Scientific insight –clinical discovery and health system performance	▪ Across-region integration ▪ Lab information integrated with electronic health record and healthcare evaluation function	The UK Department of Health & Social Care committed “£4 billion over a five-year period (2016-21) in digital technology, systems and infrastructure, to provide the health and care system with the digital capability and capacity it needs ….” [9].
**Operations**				
Entry/exit point for innovation	Technology creep and poorly performing legacy technology	▪ Appropriate health technology management	▪ Open application and evaluation process ▪ Proposals accepted from all stakeholders ▪ Explicit timelines ▪ Reassessment process	NHS England, has announced its decision to revisit tests annually, and considering the co-ordinated replacement of older tests with new and emerging approaches, including considering where evidence still needs to be collected to validate the benefit of moving to [whole-genome sequencing], and identifying where alternative genomic diagnostics, such as gene panels or microarrays, will continue to be needed [9].
Evaluative Function	Avoid low value care	▪ Legitimacy in decision-making ▪ Clear signal for innovators	▪ Adherence to key principles in health technology assessment including transparency, timeliness and stakeholder engagement [10] ▪ Consistent evaluative framework	An evaluative framework for genetic testing developed for the US Department of Defense recognized the practical need to triage adoption decisions based level on urgency through the use of rapid review and real-world evaluation of new tests [11].
Service models	Inequitable and inefficient care	▪ Care coordination	▪ Across-region coordination	NHS England Genomic Laboratory Hubs [12] and US Department of Veteran’s Affairs dedicated service centres for testing [13].
Awareness and care navigation	Confusion or lack of information regarding test availability	▪ Access to care	▪ Available, up-to-date information of test availability and how to access ▪ Additional supports for care navigation	In France, where testing is more variable across regions, lists of different laboratory sites with contact information are provided [14].
**Healthcare Environment**				
Integration of Innovation and Healthcare Delivery	Care lagging behind pace of care innovation and scientific advances	▪ Maximize care value	▪ Private public sector partnerships, and/or ▪ Integration of investigational and established technology	UK and Australian private-public-sector partnerships [15,16]. In Ontario, Canada, reflex testing for newly diagnosed cases of NSCLC (adenocarcinoma/non-squamous) uses a panel consisting of established and investigational biomarkers [17]
Financing approach	Care interruptions, wait times or unsustainable care	▪ Maximize care value ▪ Access to care ▪ Sustainable care delivery	▪ Funds available once adoption decision made ▪ Clear value-based, funding formula, amenable to reassessment ▪ Funding for test development, additional human resource costs considered	The US Centers for Medicare and Medicaid Services (CMS) have attempted to incentivize molecular diagnostic innovation by enabling manufacturer-set free pricing for FDA-cleared or approved tests under certain conditions [18].
Education and Training	Inappropriate care; medical error; care lagging behind pace of care innovation	▪ High quality workforce and care delivery	▪ Training that addresses continuing professional development, knowledge transfer and quality improvement ▪ Across-region educational standards	The Genomics Education Programme (GEP) in England plans to develop “genomic competencies for specialty training”, human resource planning, and providing supports for “curricula development and medical revalidatio” [9].
Regulation	Substandard care , negligence and legal liability	▪ Minimize preventable harm to individuals from poor test quality	▪ Regulation that addresses human resource qualifications and training, documentation of records, quality control processes, and proficiency testing [19,20,21]. ▪ Across-region analytic standards	Regulation is typically addressed through accreditation processes that conform with the International Organization for Standardization (ISO) including ISO 15189 Medical Laboratories. Examples include regulation of clinical genetic testing through CLIA in the US and Canada
Data privacy and security	Inappropriate identification of patients and family members	▪ Minimize preventable harm to individuals and families from testing	▪ Framework that addresses privacy and security concerns from genetic testing ▪ Across-region privacy standards	The Global Alliance for Genomics & Health, has created a Framework and “Core Elements for Responsible Data Sharing” [22].

## Data Availability

Not applicable.

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
