# Peer review of "Effective and Efficient Delivery of Genome-Based Testing-What Conditions Are Necessary for Health System Readiness?"

_healthcare, 2022, doi:10.3390/healthcare10102086_

Round 1
Reviewer 1 Report
The paper is basically a critical opinion on the supply of molecular tests in health system based in a qualitative information. Despite the manuscript has acceptable language quality the punctuation needs an additional revision. Please address the following points:
1. The paper is basically a qualitative research focused in the literature research and interviews. Actually, the type of manuscript should be changed to full paper.
2. Extend the introduction section with special emphasis in how genome based testing is relevant to the diagnosis of emergent diseases. Give concrete applications related with particular diseases and compare this alternative with others already available
2. A statistical analysis should be applied to the results of the interviews in order to present qualitative results. This information is mandatory for this kind of research.
3. No figures are added to the text which decrease the interest of readers. Please include graphics from the statistical analysis.
4. There are no conclusions which is quite unusual considering that this is a qualitative research.
Author Response
The paper is basically a critical opinion on the supply of molecular tests in health system based in a qualitative information. Despite the manuscript has acceptable language quality the punctuation needs an additional revision. Please address the following points:
- The paper is basically a qualitative research focused in the literature research and interviews. Actually, the type of manuscript should be changed to full paper.
Thank you for the suggestion. We will change the manuscript to “full paper” after also modifying the paper based on your further suggestions.
- Extend the introduction section with special emphasis in how genome based testing is relevant to the diagnosis of emergent diseases. Give concrete applications related with particular diseases and compare this alternative with others already available
We have elaborated on the impact of testing at the beginning of the opening paragraph:
“They are also increasingly playing a role in the identification of disease, with comprehensive genomic testing approaches demonstrating much higher diagnostic yields than conventional approaches. A prospective study of 103 children with suspected underlying genetic disorders led to 18 new diagnoses of disease when using whole-genome sequencing compared to conventional approaches to diagnosis (i.e., targeted gene sequencing).”
- A statistical analysis should be applied to the results of the interviews in order to present qualitative results. This information is mandatory for this kind of research.
No statistical analyses were conducted on interview data as we did not use a summative or ethnographic approach to content analysis which requires it. However, the reviewer is correct to point out that some analysis of qualitative findings should be presented, such as main findings , interpretations and emerging themes (as outlined, for example in the ‘Standards for Reporting Qualitative Research:A Synthesis of Recommendations, O’Brien BC, Harris IB, Beckman TJ, et al. Standards for Reporting Qualitative Research: A Synthesis of Recommendations. Academic Medicine 2014;89:1245–51. doi:10.1097/ACM.0000000000000388). As such we have included much more statistical descriptive information on interviewees and an analysis of responses given.
- No figures are added to the text which decrease the interest of readers. Please include graphics from the statistical analysis.
Thank you for this suggestion. We have added two tables to the manuscript which include the thematic analysis and interviewee characteristics.
- There are no conclusions which is quite unusual considering that this is a qualitative research.
Thank you for the suggestion. Conclusions have been added.
Reviewer 2 Report
In my opinión, another point to assess, not considered, I believe, would be the development of legislation on this type of test, and especially on the restriction on the use of its results, to guarantee patient privacy.
On the other hand, it would be recommendable also de education and well results transmission of the information to the patient in an adequate way.
Author Response
In my opinión, another point to assess, not considered, I believe, would be the development of legislation on this type of test, and especially on the restriction on the use of its results, to guarantee patient privacy.
Thank you for the comment. We have tried to address this in Section 3.3.5 (Data Privacy and Security) through the following:
“Testing also raises ethical issues, such as the duty to warn first-degree relatives who may have a high chance of carrying a disease-causing gene.[80] As such, data requirements associated with genetic testing raise privacy and security concerns that may require re-visiting of historical legislation or policies.”
On the other hand, it would be recommendable also de education and well results transmission of the information to the patient in an adequate way.
Thank you for the suggestion. This is an important point and it has now been added to section 3.3.5 (Data Privacy and Security) :
Data privacy and security concerns may also be addressed through education and training (see Section 3.3.3). A core competency framework developed by NHS Health Education England, for example, identified six areas of proficiency for those responsible for communicating test results. A part of the framework addresses appropriate commu-nication of genomic results, including understanding “the implications of genomic testing for insurance, including the UK Code on Genetic Testing and Insurance”[87]
Round 2
Reviewer 1 Report
N/A